# The effects of vertical trunk supportability improvement on one-leg rebound jump efficiency

Kinoshita Kazuaki[ID][1]◎*, Kazunari Ishida[2]◎, Masashi Hashimoto[3], Hidetoshi Nakao[4], Yuichiro Nishizawa[2], Nao Shibanuma[2], Masahiro Kurosaka[2], Shingo Otsuki[5]◎

1 Department of Physical Therapy, Faculty of Rehabilitation, Shijonawate Gakuen University, Osaka, Japan, 2 Department of Orthopaedic Surgery, Kobe Kaisei Hospital, Kobe, Japan, 3 Faculty of Health Sciences, Naragakuen University, Nara, Japan, 4 Faculty of Rehabilitation, Osaka Kawasaki Rehabilitation University, Osaka, Japan, 5 Department of Sport and Health Science, Osaka Sangyo University, Osaka, Japan

◎ These authors contributed equally to this work.
* k-kinoshita@reha.shijonawate-gakuen.ac.jp

**Data Availability Statement:** This data is related to privacy such as personal information. It is not possible to share data from third parties freely. The data underlying the results presented in the study

## Abstract

The purpose of this study was to examine the effects of vertical trunk supportability improvement on activities. The study participants were 36 people. Trunk function and physical performance were evaluated using the following tests: trunk righting test (TRT), maximal isometric knee extensor strength test, side hop test, triple hop distance test, stabilometry of one-leg standing, and one-leg rebound jump. The measurement was performed three times: pre-trunk training (pre), post-trunk training for 1 month (post), and 2 months after the second measurement (detraining). Details of trunk training: Two sets of 30-s maximal lateral reach exercises on each side, following the four sets of 15-s maximal raising trunk exercises on each side. The results with TRT in post-training were significantly larger than those in pre-and detraining. Similarly, the results with one-leg rebound jump efficiency in post-training were significantly larger than those in pre-and detraining. Our findings indicate that jump efficiency changes in proportion to the change in vertical trunk supportability.

## Introduction

Trunk stability involves many muscles, and these muscles are classified into local and global muscles [1]. The local muscles comprise muscles that insert into or have their origin in the lumbar vertebrae, and the global muscles comprise muscles that have their origin in the pelvis and are inserted into the thorax. From the structure, local muscles are important for the segmental stability between each vertebra. Hodges et al. reported that the feed-forward control of the local muscles occurs before any motion [2]. They also reported that the response of the trunk muscles is altered according to the time of upper limb movement for stable control of the trunk muscles [3]. In other words, the relationship between the trunk and motion must be such that the position and movement of the trunk can be controlled so that it can optimally produce, transmit, and control the force and motion at extremities [4, 5].

are available from the ethics committee of Shijonawate Gakuen University,+8172-863-5043. Requests for data access may also be sent to reha@shijonawate-gakuen.ac.jp for approval.

**Funding:** The authors received no specific funding for this work.

**Competing interests:** The authors have declared that no competing interests exist.

The importance of trunk function in activities such as running and jumping has been widely reported [2, 6–9]. Butcher et al. found that both trunk stability and leg strength training, but not trunk stability only or leg strength training only, further increased the vertical takeoff velocity between the third- and ninth-week testing periods in athletes [6]. According to Leetun et al., hip and trunk weakness in female athletes reduces their ability to stabilize their hip and trunk more than male athletes. The results suggested that female athletes may move their hip joints and trunk more than male athletes, which is associated with noncontact injuries [7]. Melegati et al. divided soccer players into an intervention group that performed trunk stability exercises and a control group that performed lower extremity strength exercises. The intervention group had a 39.71% reduction in the injury incidence rate compared to the control group. The reasons for this result are dynamic trunk control, dynamic and static balance, neuromuscular control, and improved flexibility and strength of the major muscle groups of the pelvic girdle [9]. These reports suggest that improving trunk function contributes to improving performance and preventing injury. Therefore, trunk function training is often provided in the field.

Most trunk function training methods are performed with the patient in the supine position. However, when the importance of the core in sports performance was examined in athletes, the results were not very promising [10–12]. One possible reason for this discrepancy is that the core test is not specific to motor skills [13]. Considering that the trunk receives gravity in a vertical direction intermittently, a static muscle endurance test is not an accurate assessment to evaluate the role of the core when considering the athletic performance in a healthy athletic population. Currently, neither are there means to dynamically assess the core and its potential role in athletic performance, nor is there a test to evaluate how well the core transfers the forces [13]. Therefore, Kinoshita et al. suggested that it is important to evaluate and train the vertical trunk function in bipedal walking [14]. A novel method named the trunk righting test (TRT) was recently described to evaluate vertical trunk function [14–16]. In this method, the patient is evaluated in the sitting position, and the vertical trunk supportability is assessed by loading the vertical direction. The trunk function evaluated by a TRT is reportedly correlated with the TRT measuring side (ipsilateral side) knee extension strength, ipsilateral side dynamic balance test, and timed up and go test in patients with knee osteoarthritis [16]. This result would assess the ability of the trunk to resist gravity. It would be relevant to the evaluation and treatment of the trunk in sports movements such as running, jumping, and cutting against gravity. However, it is unclear which activities can be performed better with improvement in vertical trunk supportability, achieved through rehabilitation. This may help develop effective rehabilitation strategies. Thus, the purpose of this study was to examine the effects of vertical trunk supportability improvement on activities.

## Materials and methods

Of the 41 healthy male and female university students initially selected for analysis, we excluded those with neurological or orthopedic abnormalities and those who complained of pain during the measurement were judged to be at risk of participating in the measurement. These 41 individuals were chosen randomly. The remaining 36 (16 males, 20 females; mean age, 19.7 ± 0.7 years; mean height, 164.0 ± 8.0 cm; mean weight, 60.5 ± 11.7 kg) were enrolled as study participants, and all participants were measured bilaterally. All participants were students who did not participate in sports on a regular basis. All participants understood the purpose of this study and provided written informed consent prior to participation according to the ethical standards of the Declaration of Helsinki. The study protocol was approved by the ethics committee of Shijonawate Gakuen University (approval number: 20–7).

Trunk function and physical performance were evaluated using the following tests: TRT [15, 16], maximal isometric knee extensor strength test [17], side hop test [18], triple hop distance test [19], stabilometry of one-leg standing [20], and one-leg rebound jump.

The TRT was performed as described in a previous report [15, 16]. Briefly, the subjects were seated in a box with their feet above the ground. The subject was in a sitting position with his shoulders moved 10 cm outward from the median sitting position, and the sensor pad was fixed on the inner part of the acromioclavicular joint by adjusting the length of the belt restraint strap to be perpendicular to the bearing surface. The subjects applied maximum power to the belt for 5 s, and the highest values were measured using a handheld dynamometer (μTas F-1; ANIMA Co., Tokyo, Japan; Fig 1). The test was conducted with a posture mirror in front and instructed that the line connecting both shoulders was parallel to the ground. The measurements were repeated thrice with at least a >30-s interval to negate the influence of fatigue. The patients were confirmed to have no fatigue. The mean of the three measurements was normalized by dividing the body weight. Measurements were performed bilaterally. This test has been reported to have an intraclass correlation coefficient of over 0.90 and can be performed with excellent reproducibility with the same examiner [15].

The maximum isometric knee extensor strength test was performed using a handheld dynamometer. The participant was instructed to sit in a median position and place both hands on the upper limbs to prevent compensation. The handheld dynamometer strap was secured to a standardized attachment on the couch leg. The length of the strap allows for isometric contraction with the knee at 90˚ during extension. The handheld dynamometer was positioned vertically in front of the tibia at the center of the medial ankle. The maximum isometric knee extensor strength was measured thrice. The average of the three measurements was normalized by dividing the weight, and that value was used in this study. This test has been reported to have an intraclass correlation coefficient of over 0.92 and can be performed with excellent reproducibility within the same examiner [17].

The side hop test was performed according to a previous report [18]. Briefly, the subjects jumped between 30 cm and one leg 10 times quickly. The time was measured three times, and the average value was used for the evaluations. If the player stepped on the line or missed the balance, the experiment was repeated. This test has been reported to have an intraclass correlation coefficient of 0.84 and can be performed with excellent reproducibility [21].

The triple hop distance test measured the distance after three consecutive jumps forward on a single leg. The jumps were required to perform with the ipsilateral legs and finish with a single-leg landing. The measurement was performed three times, and the average value was used as the measured value. This test has been reported to have an intraclass correlation coefficient of 0.88 and can be performed with excellent reproducibility [22].

The measurement of stabilometry of single-leg standing, including the total locus length, area of sway, locus length per unit area, and center of pressure speed, was performed using a center of gravity sway meter (G-5500; ANIMA Co., Tokyo, Japan; Fig 2), modified from the one use by Ageberg et al. [20]. Briefly, the contralateral leg was kept in a neutral hip position with 90˚ of knee joint flexion during single-leg standing. Both upper limbs were placed in front of the chest. The subjects were instructed to look and focus on one point 65 cm ahead. The measurements were performed three times for 10 s, and the average value was measured. The subjects practiced the maneuver before the test. A retry was performed only when the posture collapsed. The locus length per unit area was calculated by dividing the total locus length by the area of sway. This test has been reported to have an intraclass correlation coefficient of 0.68–0.83 and can be performed with excellent reproducibility [20].

The one-leg rebound jump was measured using a floor reaction force meter (AMTI, Inc., USA, Massachusetts, Watertown) to determine the jump height and jump efficiency as

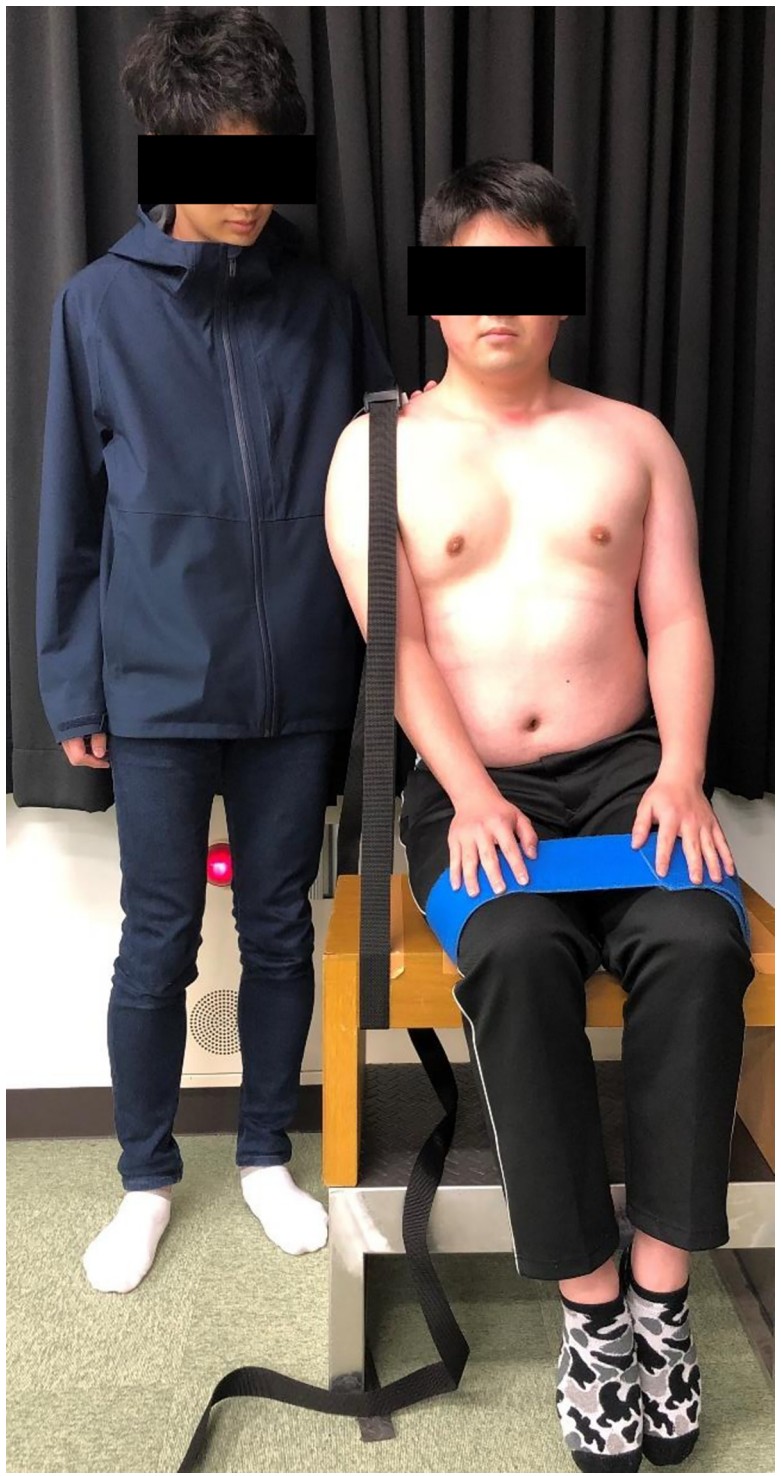

**Fig 1. Trunk righting test.**

modified by Kariyama et al. [23]. The floor reaction force meter was measured at a sampling frequency of 1000 Hz. Briefly, jumps consisted of four repeated rebound jumps in the vertical direction with a one-leg takeoff from a standing posture. During the jump, the subjects were

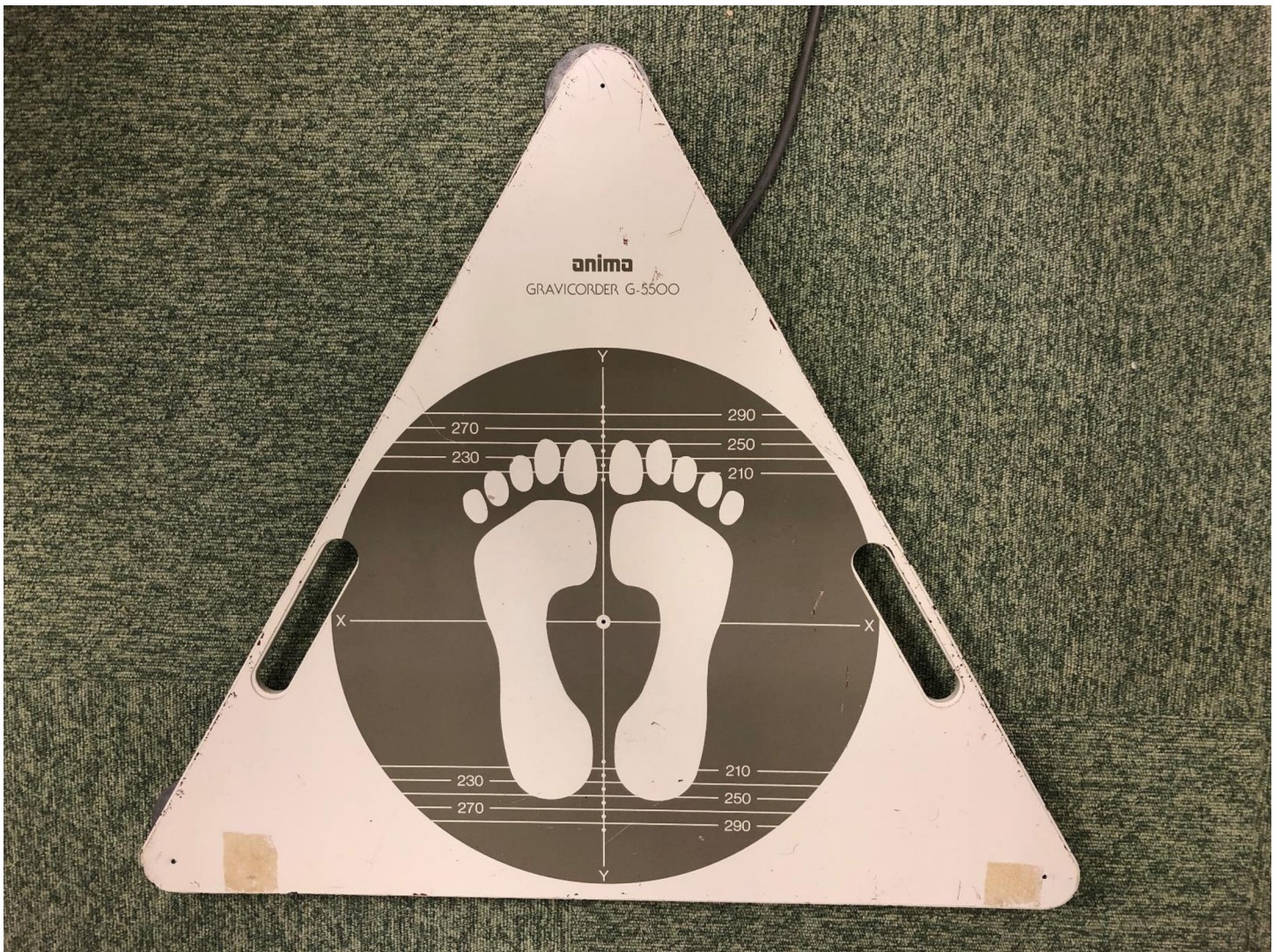

**Fig 2. Force platform.**

instructed to keep the contact time as short as possible and to jump as high as possible during the fourth jump. The fourth measurement was used for the evaluation (Fig 3). The same directed two-legged jump test by Kariyama et al. showed an intraclass correlation coefficient of 0.97. This study was reduced with one leg [23].

The flight time and railroad crossing time were measured based on the floor reaction force data. The flight time was defined as the time from the foot takeoff when the floor reaction force returned to the baseline, to foot landing when the floor reaction force increased again. The time from foot landing when the floor reaction force rose again from the baseline to the foot release when the floor reaction returned to the baseline was defined as the railroad crossing time.

Jump efficiency was calculated using the quotient of the jump height and railroad crossing time. The jump height was calculated based on the obtained flight time (= $1/8 \times$ gravity acceleration $\times$ flight time$^2$) [24]. A gravitational acceleration of 9.81 m/s$^2$ was used for the calculation. The measurement was performed twice, and a larger jump height was used as the result.

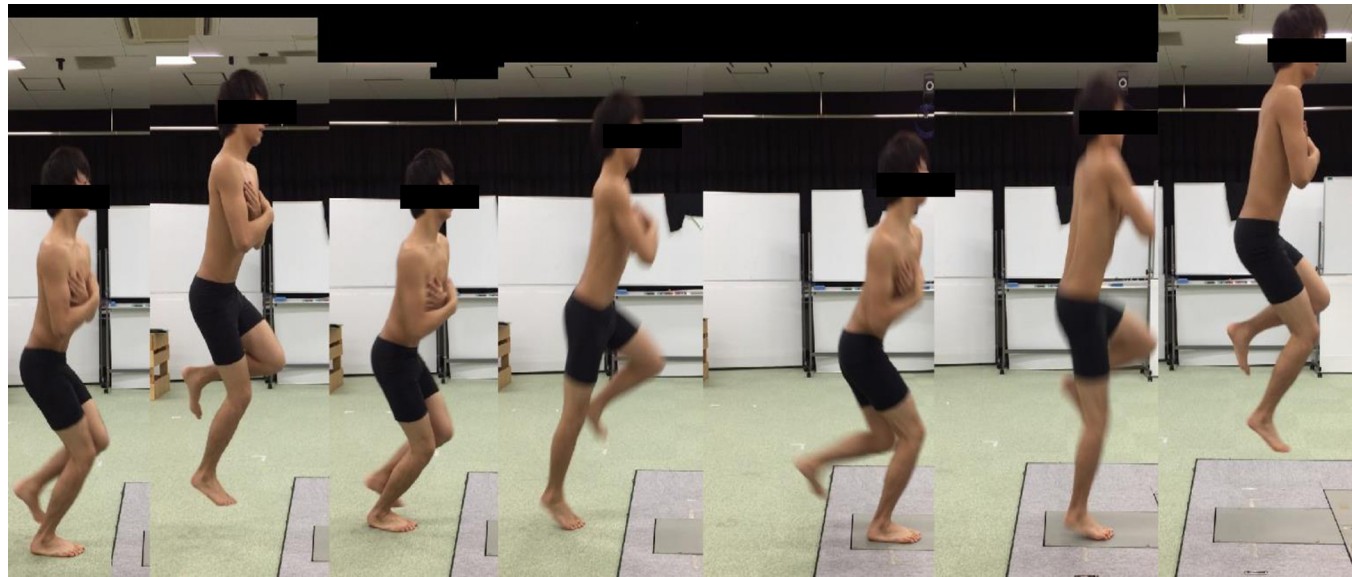

**Fig 3. One leg rebound jump.** Jumps consisted of four repeated rebound jumps in the vertical direction with a one-leg take-off from a standing posture. For rebound jump, the subjects were instructed to keep the contact time as short as possible and to jump as high as possible during the fourth jump. The fourth measurement was used.

The measurement was performed three times: pre-trunk training (pre), post-trunk training for 1 month (post), and 2 months after the second measurement (detraining). Details of trunk training: Two sets of 30-s maximal lateral reach exercises on each side, following the four sets of 15-s maximal raising trunk exercises on each side (Fig 4). Butcher et al. performed basic trunk stability exercises with little external load in the first phase of trunk stability training, and the method involved three sets of five repetitions. Each posture was held for 5 s [6]. Melegati's research showed that the training was repeated four times for 20 s on each side [9]. The intervention was determined based on these reports and the subjects' implementation of the program. Trunk training was conducted three times a week, one of which was conducted with all participants to check the training and implementation status. One month (4 weeks) of short trunk training was conducted for the nervous system, such as muscle recruitment, rate coding, and synchronization.

Statistical analysis was performed using the SPSS statistical program (version 21, IBM Corporation). Based on the sample size calculated with the G*power 3.1.9 program ($\alpha$ level 0.05, power 0.80, and estimated effect size 0.25), the total sample size required was 29. Therefore, we included 36 subjects, and the post hoc analysis for one-way repeated measures ANOVA further confirmed that the power is 0.727. Before using statistics, the normal distribution of data was confirmed using the Shapiro-Wilk test. Parametric tests were conducted using one-way repeated measures ANOVA of variance, followed by multiple comparisons by the Bonferroni test. Nonparametric tests were conducted using the Friedman test, followed by multiple comparisons by the Bonferroni test. Statistical significance was set at $p < 0.05$.

## Results

The results are presented in Fig 5. There were significant differences among the TRT, maximal isometric knee extensor strength test, side hop test, jump height, jump efficiency, and center of pressure speed. The results with TRT in post-training were significantly larger than those in pre-and detraining. Similarly, the results with one-leg rebound jump efficiency in post-

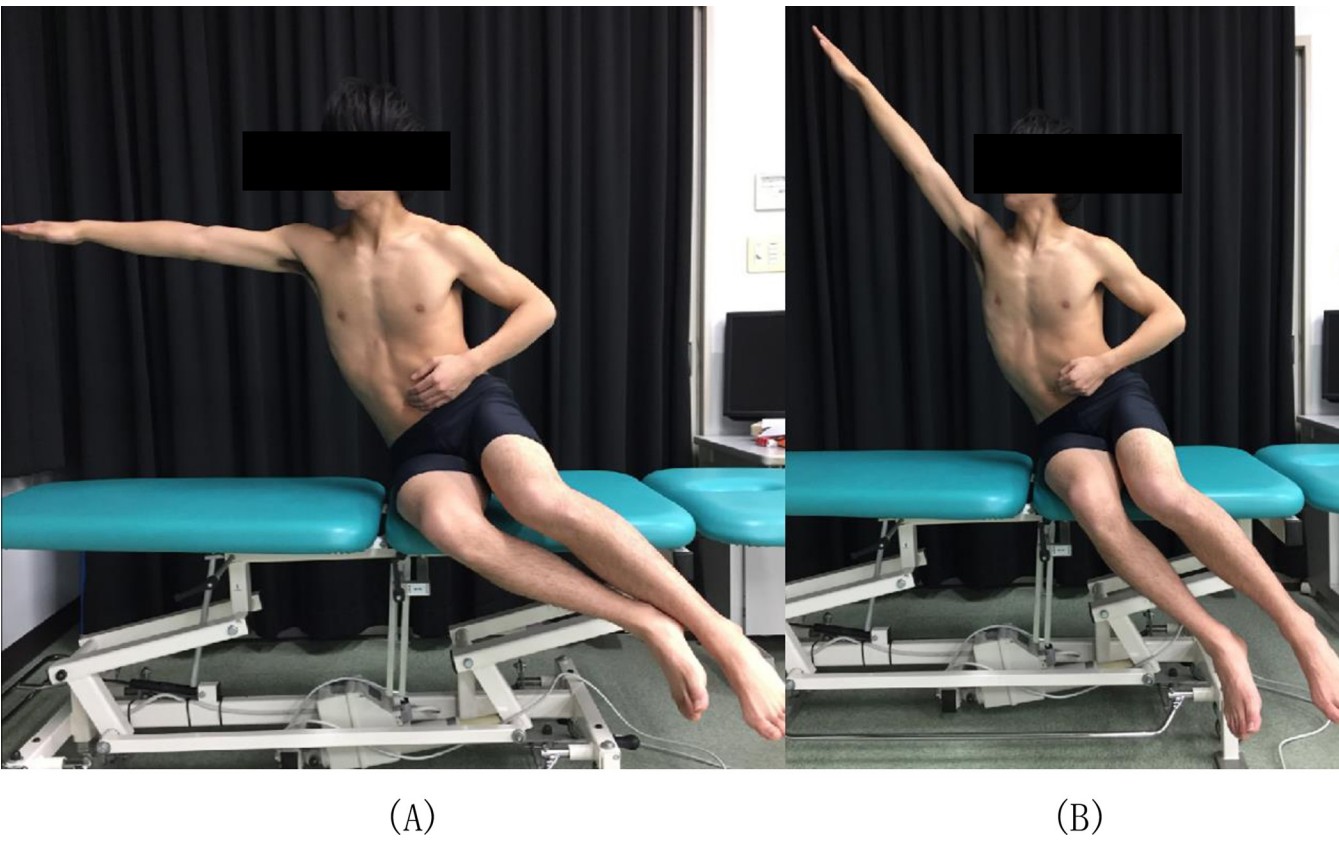

**Fig 4. Trunk training.** Subjects had two sets of 30 second maximal lateral reach exercises on each side (A) and the second had four sets of 15 second maximal raising trunk exercises on each side (B).

training were significantly larger than those in pre-and detraining. On the other hand, the results in post-training with maximal isometric knee extensor strength test were significantly larger, and with the side hop test significantly faster, than the results in pre-testing; however, these results were comparable with those of detraining. The results of the triple hop distance test and the stabilometry of leg standing were comparable between the tree measurements, except for the results of the center of pressure speed (which significantly reduced following detraining).

## Discussion

The most important finding of the present study is that the one-leg rebound jump improved after trunk training, and the effects did not persist after the training was discontinued. The results suggested that the improvement in the short ground contact time of the foot during a rebound jump is caused by the improvement in vertical trunk supportability. Therefore, jump efficiency changes with the change in vertical trunk supportability.

Hsu et al. have shown that symmetric trunk movement in the vertical direction is significantly lower in patients with ankle fracture than in healthy subjects [25]. Matsumoto et al. reported that weakness of the lower trunk muscles could lead to poor control of vertical acceleration of the center of gravity during the loading and mid-stance phases of the gait cycle [26]. Furthermore, Kinoshita et al. reported that physical functions are correlated with vertical trunk supportability evaluated by TRT in patients with knee OA, suggesting that healthcare

| | | Pre | Post | Detraining | $p$ |
|---|---|---|---|---|---|
| Trunk righting test | N/kg | 3.2 ± 0.9 | 3.7 ± 0.7 | 3.4 ± 0.8 | $p = 0.00$ |
| | | | ** | ** | |
| Maximal isometric knee extensor strength test | N/kg | 3.7 ± 0.9 | 4.1 ± 1.0 | 4.0 ± 1.1 | $p = 0.00$ |
| | | | ** | | |
| | | | | * | |
| Side hop test | S | 11.8 ± 4.1 | 11.2 ± 4.4 | 10.4 ± 3.2 | $p = 0.00$ |
| | | | ** | | |
| | | | | ** | |
| Triple hop distance test | cm | 409.9 ± 81.5 | 421.3 ± 82.6 | 413.4 ± 91.6 | $p = 0.19$ |
| One-leg rebound jump | jump height | m | 0.16 ± 0.04 | 0.17 ± 0.05 | 0. 15 ± 0.05 | $p = 0.00$ |
| | | | | | ** | |
| | | | | ** | | |
| | jump efficiency | m / s | 0.51 ± 0.18 | 0.57 ± 0.22 | 0.49 ± 0.18 | $p = 0.00$ |
| | | | | ** | ** | |
| Stabilometry of one leg standing | total locus length | cm | 39.0 ± 9.8 | 38.2 ± 10.5 | 37.9 ± 9.0 | $p = 0.08$ |
| | area of sway | cm$^2$ | 1.8 ± 0.6 | 1.9 ± 0.8 | 1.9 ± 0.7 | $p = 0.99$ |
| | locus length per unit area | 1/cm | 23.0 ± 5.1 | 22.6 ± 7.7 | 22.1 ± 6.9 | $p = 0.07$ |
| | center of pressure speed | cm/s | 3.9 ± 1.0 | 3.8 ± 1.0 | 3.8 ± 0.9 | $p = 0.03$ |
| | | | | ** | | |

**The result of the measurement** *: $p < 0.05$ , **: $p < 0.01$

**Fig 5. The result of the measurement.**

workers must consider trunk function as well as lower extremity function to improve physical function. These results suggest that the decrease in vertical trunk supportability leads to a decrease in lower limb load; thus, vertical trunk supportability is strongly related to lower limb load.

Jump efficiency is important for high-velocity concentric and eccentric muscular contractions involving the muscular stretch-shortening cycle. Furthermore, plyometric training is important for these improvements. Balance and stability during plyometric training regulate proper muscle contraction before landing. They were adjusted in a feed-forward manner [27]. Therefore, jump efficiency is important for activities. These results suggest that vertical trunk supportability is important in bipedal animal activity and is useful in longitudinal studies. The strength of the lower limbs can be considered to improve jump efficiency. The results of this study showed that the maximal isometric knee extensor strength test of the lower limb strength index increased significantly from pre-to post-intervention. However, there was no significant change from post-detraining. Therefore, this result is not considered a mere increase or decrease in lower limb strength. The results of this study support the findings of Butcher et al., who reported that 9 weeks of trunk stability training was as effective in increasing the vertical takeoff speed as leg strength training or a combination of trunk stability and leg strength training [6]. Mills et al. studied vertical jump heights after 10 weeks of different trunk training and found that the local stability muscle training group improved significantly after training, while the global mobility muscle training group did not [28]. These reports indicate that jumping ability improves regardless of the lower limb strength. However, there was no correlation

between trunk stability score and vertical jump heights [28]. Therefore, it is important to note that trunk stability does not determine vertical jump heights.

The findings of several studies support those of this study. Trunk stability training has the potential to optimize the ability of the leg muscles to produce force, because it provides a stable base for the leg muscles to contract and it strengthens the neural drive [29, 30]. Butcher et al. reported that the improvement in the vertical takeoff speed after the three-week trunk stability training was due to changes in neuromuscular control or movement patterns rather than changes in muscle structure [6, 30]. This study is similar to previous studies in that the core training period was short and the load was low. Lower limb strength results were also unaffected. A rebound jump generates a large ground reaction force upon landing, so the trunk has a large vertical impact after ground contact [31, 32]. Therefore, the abdominal muscles need to be active in order to stabilize the trunk in preparation for landing and to control the position of the trunk prior to ground contact [33, 34]. The muscles in the lower back work to control the position of the trunk after ground contact and to change the direction from a downward motion to an upward motion [34]. These studies show that the center of gravity of the trunk may lead to the next activity being controlled quickly by the lower limbs. In other words, it is inferred that improvement of core function becomes easier to control to the next activity by changes in neuromuscular control or movement pattern.

Although the side hop test showing left-right agility was significantly improved from pre-to post-training, there was no significant change from post- to detraining. The side hop test has a different course than jump efficiency. This is because the side hop test is difficult, and many of the test subjects performed it for the first time. Scinicarelli et al. examined the reproducibility of the side hop test at seven days [35]. They reported good reproducibility, but significant improvement on the non-dominant side. It is likely that the complexity of performing the side hop test plays an important role in the scores obtained. They also reported that the learning effect improved the performance in the second test session as the test subjects became more familiar with the test execution and were able to perform it faster. Motor learning was performed by repeating the side hop test. Thus, the side hop test gradually improved.

This study had three limitations. First, there was no control group. A comparison between the intervention and non-intervention groups would have strengthened our results. Second, the influence of each individual's life during the 3 months remains unknown. This may be associated with an improved maximal isometric knee extensor strength test performance. Finally, other factors that affect jump efficiency were not fully evaluated.

## Conclusions

This study examined the effects of vertical trunk supportability improvement on the ability to perform activities. The most important finding of the present study is that the one-leg rebound jump improved after trunk training, but its effects this improvement did not last after the training was discontinued. We conclude that the improvement in the short ground contact time of the foot during a rebound jump is caused by the improvement in vertical trunk supportability. Our findings indicate that jump efficiency changes in proportion to the change in vertical trunk supportability.

## Acknowledgments

We would like to express our sincere gratitude to the study participants and the past and present members of my laboratory. We would like to thank Editage (www.editage.com) for English language editing.

## Author Contributions

**Conceptualization:** Kinoshita Kazuaki, Masashi Hashimoto.

**Data curation:** Kinoshita Kazuaki.

**Formal analysis:** Kinoshita Kazuaki, Hidetoshi Nakao.

**Investigation:** Kinoshita Kazuaki, Shingo Otsuki.

**Methodology:** Kinoshita Kazuaki, Shingo Otsuki.

**Project administration:** Masahiro Kurosaka, Shingo Otsuki.

**Resources:** Kinoshita Kazuaki, Shingo Otsuki.

**Software:** Kinoshita Kazuaki, Hidetoshi Nakao.

**Supervision:** Masahiro Kurosaka, Shingo Otsuki.

**Validation:** Yuichiro Nishizawa, Nao Shibanuma.

**Visualization:** Kinoshita Kazuaki, Kazunari Ishida.

**Writing – original draft:** Kinoshita Kazuaki, Kazunari Ishida.

**Writing – review & editing:** Kinoshita Kazuaki, Kazunari Ishida.

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
