## [Decision Letter · Decision Letter 0]

22 Sep 2021

PONE-D-21-14225The effects of vertical trunk supportability improvement on one-leg rebound jump efficiencyPLOS ONE

Dear Dr. Kazuaki,

Thank you for submitting your manuscript to PLOS ONE. After careful consideration, we feel that it has merit but does not fully meet PLOS ONE’s publication criteria as it currently stands. Therefore, we invite you to submit a revised version of the manuscript that addresses the points raised during the review process.

Dear Authors, two expert in the field revised your manuscript reporting several major issues including Methods and results.

We look forward to receiving your revised manuscript.

Kind regards,

Emiliano Cè

Academic Editor

PLOS ONE

Journal Requirements:

3. PLOS requires an ORCID iD for the corresponding author in Editorial Manager on papers submitted after December 6th, 2016. Please ensure that you have an ORCID iD and that it is validated in Editorial Manager. To do this, go to ‘Update my Information’ (in the upper left-hand corner of the main menu), and click on the Fetch/Validate link next to the ORCID field. This will take you to the ORCID site and allow you to create a new iD or authenticate a pre-existing iD in Editorial Manager. Please see the following video for instructions on linking an ORCID iD to your Editorial Manager account: https://www.youtube.com/watch?v=_xcclfuvtxQ"

4. Thank you for submitting the above manuscript to PLOS ONE. During our internal evaluation of the manuscript, we found significant text overlap between your submission and the following previously published works.

- https://www.jstage.jst.go.jp/article/jpts/31/3/31_jpts-2018-388/_pdf/-char/en

- https://www.frontiersin.org/articles/10.3389/fphys.2019.01462/full

- https://www.jospt.org/doi/pdfplus/10.2519/jospt.2007.2331

We would like to make you aware that copying extracts from previous publications, especially outside the methods section, word-for-word is unacceptable, even for works which you authored. In addition, the reproduction of text from published reports has implications for the copyright that may apply to the publications.

Please revise the manuscript to rephrase the duplicated text, cite your sources, and provide details as to how the current manuscript advances on previous work. Please note that further consideration is dependent on the submission of a manuscript that addresses these concerns about the overlap in text with published work.

Reviewers' comments:

Reviewer's Responses to Questions

**Comments to the Author**

1. Is the manuscript technically sound, and do the data support the conclusions?

Reviewer #1: Yes

Reviewer #2: Partly

2. Has the statistical analysis been performed appropriately and rigorously? 

Reviewer #1: I Don't Know

Reviewer #2: Yes

3. Have the authors made all data underlying the findings in their manuscript fully available?

Reviewer #1: No

Reviewer #2: Yes

4. Is the manuscript presented in an intelligible fashion and written in standard English?

Reviewer #1: Yes

Reviewer #2: No

5. Review Comments to the Author

Reviewer #1: The duration of trunk training is not clear.

The characteristics of participants are missing.

Statistical analysis is not explained.

The stabilometry measure is not well explained, Why the authors do not used a stabilometric force platform?

In my opinion in the discussion should be more references, especially in this affirmation "Therefore, jump efficiency

is important for activities. These results suggest that vertical trunk supportability is important in bipedal animal activity and is useful in longitudinal studies. The strength of the lower limbs can be considered to improve jump efficiency. The results of this study showed that the maximal isometric knee extensor strength test of the lower limb strength index increased significantly from pre-to post-intervention. However, there was no significant change from post-detraining. Therefore, this result is not conside 195 red a mere increase or decrease in lower limb strength". In this affirmation ther is not refence "Lower limb strength results were also unaffected. The rebound jump has a greater centrifugal force on the trunk than on the lower limbs. The center of gravity of the trunk may lead to the next activity by being controlled quickly on the lower limbs. In other words, it is inferred that improvement of core function becomes easier to control to the next activity by changes in neuromuscular control or movement pattern" is necessary more references in the discussion there few and most are very old.

Reviewer #2: I would like to thank the journal for the opportunity to review this work. The authors presented data investigating the effect of trunk training on various functional testing. However, I am concerned about the study design, statistical analyses, and presentation of the data in the introduction, methods, and discussion sections. I have several minor comments offered below. Thank you.

Specific comments to author about the manuscript:

Line 37-43: Please add more specific explanations on trunk function of anatomical (muscles) and biomechanical perspectives during movement. Sentences within the first paragraph are vague and not informative, and further the authors did not provide sufficient data for a rational of this study.

Line 44-55: The authors mainly described a trunk righting test in the second paragraph, however, the authors did not explain how this test’s outcome can contribute to results of other tests, and how the trunk tithing test is associated other functional tests and further activities of daily living. It seems the authors did not provide thorough literature review in this area, and they need to add explanations why this study is important (a rational of this study)?

Line 44-55: No explanation on the intervention dose (a duration, frequency, set/reps, types, etc.) from previous studies. Thorough literature review is required, and should provide a sentence how this study bridges a gap from a current literature.

Line 37: what activities?

Line 41: what injury, what sports, what age? How can core stability be defined in that study?

Line 57-151: Authors should provide a validity and reliability scores of all testing. They should add whether testing protocols were adapted from previous studies or not. If not, please add more pictures regarding testing procedures to better understand how the procedures are done.

Line 166: As authors mentioned in the limitations of the study, not having a control group in the intervention study was a major concern. We can’t conclude whether pre-post changes were due to the intervention or other confounding factors.

Line 169-217: In the discussion section authors should add more relevant citations for each sentence. Some sentences are required supporting evidence, but it’s missing.

6. PLOS authors have the option to publish the peer review history of their article (what does this mean?). If published, this will include your full peer review and any attached files.

Reviewer #1: No

Reviewer #2: No

---

## [Author Response · Author response to Decision Letter 0]

8 Mar 2022

The author sincerely appreciates the opportunity to improve the manuscript that has been provided by the reviewer. 

The manuscript has been revised. Please review the manuscript again.

We have included the details in the "Response to reviewers".

Thank you very much for your cooperation.

---

## [Decision Letter · Decision Letter 1]

25 Mar 2022

PONE-D-21-14225R1

The effects of vertical trunk supportability improvement on one-leg rebound jump efficiency

PLOS ONE

Dear Dr. Kazuaki,

Thank you for submitting your manuscript to PLOS ONE. After careful consideration, we feel that it has merit but does not fully meet PLOS ONE’s publication criteria as it currently stands. Therefore, we invite you to submit a revised version of the manuscript that addresses the points raised during the review process.

ACADEMIC EDITOR: Dear Authors, one expert in the field re-evaluated your manuscript still reporting some minor points you should consider while revising it.

We look forward to receiving your revised manuscript.

Kind regards,

Emiliano Cè

Academic Editor

PLOS ONE

Journal Requirements:

Reviewers' comments:

Reviewer's Responses to Questions

**Comments to the Author**

1. If the authors have adequately addressed your comments raised in a previous round of review and you feel that this manuscript is now acceptable for publication, you may indicate that here to bypass the “Comments to the Author” section, enter your conflict of interest statement in the “Confidential to Editor” section, and submit your "Accept" recommendation.

Reviewer #1: All comments have been addressed

2. Is the manuscript technically sound, and do the data support the conclusions?

Reviewer #1: Yes

3. Has the statistical analysis been performed appropriately and rigorously? 

Reviewer #1: Yes

4. Have the authors made all data underlying the findings in their manuscript fully available?

Reviewer #1: Yes

5. Is the manuscript presented in an intelligible fashion and written in standard English?

Reviewer #1: Yes

6. Review Comments to the Author

Reviewer #1: Line 42 In this phrase "Trunk stability involves many muscles, and these muscles are classified into local and

43 global muscles [32]" it should be reference [1]. References should be listed in numerical order, and in the same order in which they are cited in text per example. 1,2,3,4,etc.

7. PLOS authors have the option to publish the peer review history of their article (what does this mean?). If published, this will include your full peer review and any attached files.

Reviewer #1: **Yes: **Rosa Cabanas-Valdés

---

## [Author Response · Author response to Decision Letter 1]

30 Mar 2022

The author sincerely appreciates the opportunity to improve the manuscript that has been provided by the academic editor and reviewer.

I understand you are occupied at the moment, but please review again .

---

## [Decision Letter · Decision Letter 2]

11 Apr 2022

The effects of vertical trunk supportability improvement on one-leg rebound jump efficiency

PONE-D-21-14225R2

Dear Dr. Kazuaki,

We’re pleased to inform you that your manuscript has been judged scientifically suitable for publication and will be formally accepted for publication once it meets all outstanding technical requirements.

Kind regards,

Emiliano Cè

Academic Editor

PLOS ONE

Additional Editor Comments (optional):

Reviewers' comments:

Reviewer's Responses to Questions

**Comments to the Author**

1. If the authors have adequately addressed your comments raised in a previous round of review and you feel that this manuscript is now acceptable for publication, you may indicate that here to bypass the “Comments to the Author” section, enter your conflict of interest statement in the “Confidential to Editor” section, and submit your "Accept" recommendation.

Reviewer #1: All comments have been addressed

2. Is the manuscript technically sound, and do the data support the conclusions?

Reviewer #1: Yes

3. Has the statistical analysis been performed appropriately and rigorously? 

Reviewer #1: Yes

4. Have the authors made all data underlying the findings in their manuscript fully available?

Reviewer #1: Yes

5. Is the manuscript presented in an intelligible fashion and written in standard English?

Reviewer #1: Yes

6. Review Comments to the Author

Reviewer #1: Thanks you for your corrections. I am agree with the addition of this reference "Cabrejas C, Solana-Tramunt M, Morales J, Campos-Rius J, Ortegón A, Nieto Guisado A, et al. The Effect of Eight-Week Functional Core Training on Core

Stability in Young Rhythmic Gymnasts: A Randomized Clinical Trial. Int J Environ Res Public Health. 2022; 19: 3509.

7. PLOS authors have the option to publish the peer review history of their article (what does this mean?). If published, this will include your full peer review and any attached files.

Reviewer #1: **Yes: **Rosa Cabanas-Valdés

---

## [Editor Report · Acceptance letter]

22 Apr 2022

PONE-D-21-14225R2 

The effects of vertical trunk supportability improvement on one-leg rebound jump efficiency 

Dear Dr. Kazuaki:

I'm pleased to inform you that your manuscript has been deemed suitable for publication in PLOS ONE. Congratulations! Your manuscript is now with our production department. 

Kind regards, 

on behalf of

Professor Emiliano Cè 

Academic Editor

PLOS ONE